TAS3 miR390-dependent loci in non-vascular land plants: towards a comprehensive reconstruction of the gene evolutionary history

Morozov Sergey Y. morozov@genebee.msu.su 1
Milyutina Irina A. 1
Erokhina Tatiana N. 2
Ozerova Liudmila V. 3
Troitsky Alexey V. 1
Solovyev Andrey G. 1 4
1 Belozersky Institute of Physico-Chemical Biology, Moscow State University , Moscow , Russia
2 Shemyakin-Ovchinnikov Institute of Bioorganic Chemistry, Russian Academy of Science , Moscow , Russia
3 Tsitsin Main Botanical Garden, Russian Academy of Science , Moscow , Russia
4 Institute of Molecular Medicine, Sechenov First Moscow State Medical University , Moscow , Russia
Vassetzky Yegor
Electronic publication date: 2018 Apr 16
Publication date: 2018
Volume: 6
Electronic Location ID: e4636
Received 2018 Feb 19; Accepted 2018 Mar 28
Copyright: ©2018 Morozov et al.
Copyright year: 2018
Copyright holder: Morozov et al.
License: This is an open access article distributed under the terms of the Creative Commons Attribution License, which permits unrestricted use, distribution, reproduction and adaptation in any medium and for any purpose provided that it is properly attributed. For attribution, the original author(s), title, publication source (PeerJ) and either DOI or URL of the article must be cited.
License URL: https://creativecommons.org/licenses/by/4.0/

Keywords: Silencing, Trans-acting RNA, Small interfering RNA, ARF genes, Micro RNA, Charophyte algae, Bryophytes

Funding: Russian Science Foundation 17-14-01032 Russian Foundation for Basic Research 18-04-00574-a State Assignment of MBG RAS on the base of the Unique Scientific Installation “The Fund Greenhouse” The work of Sergey Morozov, Tatiana Erokhina and Andrey Solovyev was supported by the Russian Science Foundation (grant 17-14-01032). The work of Irina Milyutina and Alexey Troitsky was supported by the Russian Foundation for Basic Research (grant 18-04-00574-a). The work of Liudmila Ozerova was supported by the State Assignment of MBG RAS on the base of the Unique Scientific Installation “The Fund Greenhouse”. The funders had no role in study design, data collection and analysis, decision to publish, or preparation of the manuscript.

==============================
Trans-acting small interfering RNAs (ta-siRNAs) are transcribed from protein non-coding genomic TAS loci and belong to a plant-specific class of endogenous small RNAs. These siRNAs have been found to regulate gene expression in most taxa including seed plants, gymnosperms, ferns and mosses. In this study, bioinformatic and experimental PCR-based approaches were used as tools to analyze TAS3 and TAS6 loci in transcriptomes and genomic DNAs from representatives of evolutionary distant non-vascular plant taxa such as Bryophyta, Marchantiophyta and Anthocerotophyta. We revealed previously undiscovered TAS3 loci in plant classes Sphagnopsida and Anthocerotopsida, as well as TAS6 loci in Bryophyta classes Tetraphidiopsida, Polytrichopsida, Andreaeopsida and Takakiopsida. These data further unveil the evolutionary pathway of the miR390-dependent TAS3 loci in land plants. We also identified charophyte alga sequences coding for SUPPRESSOR OF GENE SILENCING 3 (SGS3), which is required for generation of ta-siRNAs in plants, and hypothesized that the appearance of TAS3-related sequences could take place at a very early step in evolutionary transition from charophyte algae to an earliest common ancestor of land plants.

Introduction

Plant chromosomal loci of trans-acting small interfering RNAs (ta-siRNAs) and microRNAs (miRNAs) encode non-protein-coding and protein-coding precursor transcripts, which are synthesized by RNA polymerase II and include cap-structures and poly-(A) tails. In plants, primary miRNA transcripts forming internal imperfect hairpins are processed by a protein complex including Dicer-like protein 1 (DCL1), HYL1 and SERRATE to give RNA duplexes with 2-nucleotide 3′-overhangs, which are then terminally methylated by specific RNA methylase HEN1. One strand of such duplexes, being typically of 21 nucleotides in length and representing a mature miRNA, is selectively recruited by Argonaut (AGO) family protein to an effector complex targeting a specific RNA for AGO-mediated endonucleolytic cleavage or translational repression (Rogers & Chen, 2013; Axtell, 2013; Bologna & Voinnet, 2014; Borges & Martienssen, 2015; Chorostecki et al., 2017).

Some specific microRNAs are able to initiate production of ta-siRNAs (and other secondary phased RNAs—phasiRNAs) by an step-by-step processing of long double-stranded RNA by DCL4 from a start point defined by miRNA-directed cleavage of a single-stranded RNA precursor in a “phased” pattern. These PHAS loci include non-coding TAS genes and genes encoding penta-tricopeptide repeat-containing proteins (PPRs), nucleotide-binding and leucine-rich repeat-containing proteins (NB-LRRs), or MYB transcription factors (Allen & Howell, 2010; Zhai et al., 2011; Xia et al., 2013; Fei, Xia & Meyers, 2013; Axtell, 2013; Yoshikawa, 2013; Zheng et al., 2015; Komiya, 2017; Liu et al., 2018; Deng et al., 2018). Biogenesis of ta-siRNAs includes initial AGO-dependent miRNA binding at single or dual sites of the precursor transcripts and their subsequent cleavage. The further process is dependent on plant RNA-dependent RNA polymerase 6 (RDR6) and SGS3 proteins participating in the formation of dsRNA, which is then cleaved in a sequential and phased manner by DCL4 with assistance of DRB4 (dsRNA binding protein). The resulting ta-siRNAs (mostly of 21 bp in length), similar to miRNAs, are methylated by HEN1 protein (Allen & Howell, 2010; Axtell, 2013; Fei, Xia & Meyers, 2013; Yoshikawa, 2013; Bologna & Voinnet, 2014; Komiya, 2017; Deng et al., 2018).

Arabidopsis TAS3a transcript, first identified by Allen et al. (2005), gives rise to two near-identical 21-nucleotide tasiARFs targeting the mRNAs of some auxin-responsive transcription factors (ARF2, ARF3/ETT and ARF4). Most angiosperm TAS3 primary transcripts are recognized by miR390 and cleaved by AGO7 at the 3′ target site, whereas the 5′ miRNA target site is non-cleavable. However, the number of miR390 cleavage sites, organization of tasiARF sequence blocks and phasing registers may vary among different TAS3 genes of vascular plants (Allen & Howell, 2010; Axtell, 2013; Fei, Xia & Meyers, 2013; Zheng et al., 2015; Xia et al., 2013; Xia, Xu & Meyers, 2017; De Felippes et al., 2017; Komiya, 2017; Deng et al., 2018). Moreover, miR390 may additionally target and inhibit protein-coding gene transcripts, such as StCDPK1 related to auxin-responsive pathway (Santin et al., 2017).

Previously, we described a new method for identification of plant TAS3 loci based on PCR with a pair of oligodeoxyribonucleotide primers mimicking miR390. The method was found to be efficient for dicotyledonous plants, cycads, conifers, and mosses (Krasnikova et al., 2009; Krasnikova et al., 2011; Krasnikova et al., 2013; Ozerova et al., 2013). Importantly, at that time the structural and functional information on bryophyte TAS3 loci was available only for the model plant Physcomitrella patens (Arif, Frank & Khraiwesh, 2013), and we used our PCR-based approach as a phylogenetic profiling tool to identify relatives of P. patens TAS3 loci in 26 additional moss species of class Bryopsida and several mosses of classes Polytrichopsida, Tetraphidopsida and Andreaeopsida. Moreover, we found a putative pre-miR390 genomic sequence for an additional moss class, Oedipodipsida (Krasnikova et al., 2013). Our studies revealed that a representative of Marchantiophyta (liverwort Marchantia polymorpha, class Marchantiopsida) could also encode a candidate miR390 gene and a potential TAS3-like locus (Krasnikova et al., 2013). This finding extended the known evolutionary history of TAS3 loci to the proposed most basal land plant lineage (Ruhfel et al., 2014; Bowman et al., 2017). In addition, we sequenced putative pre-miR390 genomic locus for Harpanthus flotovianus (Marchantiophyta, class Jungermanniopsida) (Krasnikova et al., 2013). Later, our findings of TAS3-like and miR390 loci were experimentally confirmed in the studies of the transcriptomes of Marchantiophyta plants M. polymorpha (Lin et al., 2016; Tsuzuki et al., 2016) and Pellia endiviifolia (class Jungermanniopsida) (Alaba et al., 2015).

New genomic and transcriptomic sequence data for basal Viridiplantae appeared in NCBI (http://ncbi.nlm.nih.gov/sra) and Phytozome (http://www.phytozome.net) databases prompted us to perform new experimental and in silico analyses of TAS3 loci in basal taxons of Viridiplantae. In this paper, we identified previously unrecognized TAS3 loci in classes Sphagnopsida and Anthocerotopsida, as well as composite TAS6/TAS3 loci in classes Tetraphidiopsida, Polytrichopsida, Andreaeopsida and Takakiopsida. Additionally, we revealed SGS3-coding sequences in charophytes and analyzed their evolutionary links.

Materials and Methods

Dried material for Sphagnum angustifolium and S. girgensohnii were taken from herbarium at Department of Biology, Moscow State University. Total DNA was extracted from dry plants using the Nucleospin Plant Extraction Kit (Macherey-Nagel, Düren, Germany) according to the protocol of the manufacturer. For PCR amplification, the following degenerate primers were used: a forward primer Spha-TASP (5′-GGCGRTAWCCYTACTGAGCTA-3′) and reverse primer Spha-TASM (5′-TAGCTCAGGAGRGATAMMBMRA-3′). For PCR, 30 cycles were used with a melting temperature of 94 °C –3′, and the next steps are as follows: an annealing temperature 94 °C –20”, 65 °C –20”, 58 °C –30”, and an extending temperature of 72 °C followed by a final extension at 72 °C for 5′. PCR products were separated by electrophoresis of samples in a 1.5% agarose gel and purified using the Gel Extraction Kit (Qiagen, Hilden, Germany). For cloning, the PCR-amplified DNA bands isolated from gel were ligated into pGEM-T (Promega, Madison, WI, USA). The resulting clones were screened by length in 1,5% agarose gel. The plasmids were used as templates in sequencing reactions with an automated sequencer (Applied Biosystems, Foster City, CA, USA) 3730 DNA Analyzer with facilities of “Genom” (Moscow, Russia).

Sequences for comparative analysis were retrieved from NCBI (http://www.ncbi.nlm.nih.gov/), Phytozome (http://www.phytozome.net) and 1,000 Plant Transcriptome Project (“1KP”) (http://1kp-project.com/blast.html). Sequence similarities were analysed by NCBI Blast at http://blast.ncbi.nlm.nih.gov/BlastAlign.cgi. The presence of open reading frames within retrieved sequences was analysed at http://web.expasy.org/translate/. The nucleic acid sequences and deduced amino acid sequences were analyzed and assembled using the NCBI. Conserved domains in the amino acid sequences SGS3 were identified using the CD-Search of the NCBI.

The sequences of SGS3 protein and TAS3 nucleotide sequences were aligned by MAFFT version 7 software (Katoh & Standley, 2014). The phylogenetic tree was constructed by the Neighbor-Joining method with 1,000 bootstrap replications in MEGA7 (Kumar, Stecher & Tamura, 2016). The evolutionary distances were computed using the JTT matrix-based method and are in the units of the number of amino acid substitutions per site. The rate variation among sites was modeled with a gamma distribution (shape parameter = 1).

Results

TAS3 loci in Bryophyta (classes Sphagnopsida and Takakiopsida)

It is commonly accepted that mosses of classes Sphagnopsida and Takakiopsida represent most basal lineages in Bryophyta (Shaw et al., 2010; Shaw, Szövényi & Shaw, 2011; Rosato et al., 2016). Previously, using primers, which have allowed us to detect pre-miR390 and TAS3 loci in Bryopsida and some other moss classes, we failed to identify pre-miR390 and TAS3 genes in genus Sphagnum (Krasnikova et al., 2013). However, a predicted sequence of pri-miR390 from Sphagnum fallax was recently reported (Xia, Xu & Meyers, 2017). This finding prompted us to re-evaluate the occurrence of TAS3-like loci in Sphagnopsida. To this end, we designed a new pair of degenerated PCR primers Spha-TASP and Spha-TASM, which differed from those used previously (Krasnikova et al., 2011; Krasnikova et al., 2013). As a positive control, we used plasmid DNA carrying cloned TAS3 gene of Andreaea rupestris, a representative of basal Bryophyta (Krasnikova et al., 2013). Like the positive control, two total DNA probes from Sphagnum angustifolium and S. girgensohnii gave a single main PCR product of the expected size (Fig. 1). Cloning and sequencing of these PCR fragments revealed two TAS3-like primary structures having 285 (S. angustifolium) and 292 (S. girgensohnii) bases in length and exhibiting 96% identity (e-value = 2e−131). We named these loci as Sphan-285 and Sphgi-292, (Fig. 2, Fig. S1 and Table 1). Despite that the degenerate miR390-mimicking primers were used for amplification of these loci, miR390 recognition sites in TAS3 species well corresponded to genomic loci of TAS3 extracted from genomic and transcriptomic data of other plants from genus Sphagnum (Fig. 2).

Figure 1 Analysis of PCR products in 1.5% agarose gel.

Amplification of genomic DNA sequences flanked by miR390 and miR390* sites. PCR products were obtained on genomic DNAs with degenerate primers. Sphagnum angustifolium (1), Sphagnum girgensohnii (2), Andreaea rupestris (3). (M), DNA size markers including bands ranging from 100 bp to 1,000 bp with 100 bp step (Sibenzyme).

Peatmosses S. angustifolium and S. girgensohnii belong to subgenera Cuspidata and Acutifolia, respectively (Shaw et al., 2010; Shaw et al., 2016). To extend search for TAS3-like loci inside genus Sphagnum we performed bioinformatics analysis of the nucleotide sequences in databases available at NCBI (Sequence Read Archive) and Phytozome (version 12.1). Phytozome has recently released genome assembly of bog moss S. fallax (version 0.5). Bog moss belongs to subgenus Cuspidata and represents the most closely related moss to S. angustifolium (Shaw et al., 2016). BLASTN search at Phytozome allowed us to reveal a TAS3-like locus (supercontig super_37), which has 100% identity to the TAS3 locus of S. angustifolium sequenced in this study (Fig. S1 and Table 1). Unexpectedly, we found an additional TAS3-like locus in S. fallax (transcript Sphfalx0293s0011, supercontig super_293). This TAS3 locus in bog moss has 277 nucleotides in length and showed only a distant relation to the S. angustifolium TAS3 (Fig. 2, Fig. S1 and Table 1).

Figure 2 Analysis of TAS3 loci in genus Sphagnum.

(A) Multiple sequence alignment of available nucleotide sequences of TAS3-like loci from mosses of genus Sphagnum along with TAS3 loci of Takakia lepidozioides and Folioceros fuciformis. Alignment was generated at MAFFT6 program. The miR390 target sites are in yellow, and putative tasiARF-a2 site is in green; tasiAP2 site is in blue. (B) The minimal evolution phylogenetic tree based on analysis of the aligned TAS3 genes from mosses of genus Sphagnum. This tree was generated according MAFFT6 program. For full plant names and accession numbers see Table 1.

To further analyze Sphagnopsida TAS3-related loci, we used BLAST analysis of Sequence Read Archive (SRA), which is the NCBI database collecting sequence data obtained by the use of next generation sequence (NGS) technology. Assembly of sequence reads of S. recurvum (subgenus Cuspidata) retrieved by BLAST search using S.fallax sequences as queries revealed two TAS3 loci (Table 1). The first locus (Sphre-283) is 283 nucleotides in length and has 98% identity to Sphan-285. The second locus (Sphre-277) shows 98% identity to Sphfalx0293s0011 (Table 1, Fig. S1). These findings indicate that two distant TAS3 loci in species of a particular subgenus of genus Sphagnum are extremely similar.

Table 1 List of the putative TAS3 loci in Sphagnopsida and Takakiopsida.

Plant species	Locus name	Subgenus	Length	Sequence source	
Sphagnum angustifolium	Sphan-285	Cuspidata	285 nts	MF682529	
S. girgensohnii	Sphgi-292	Acutifolia	292 nts	MF682530	
S. fallax	contig super_37	Cuspidata	285 nts	SRX2120232	
S. fallax	Sphfalx0293s0011	Cuspidata	277 nts	Sphfalx0293s0011a	
S. recurvum	Sphre-283	Cuspidata	283 nts	SRX1513231	
S. recurvum	Sphre-277	Cuspidata	277 nts	SRX1513231	
S. magellanicum	Sphma-285	Sphagnum	285 nts	SRX2330962	
S. magellanicum	Sphma-286	Sphagnum	286 nts	SRX2330962	
S. palustre	Sphpa	Sphagnum	partial	SRX1516347	
S. cribrosum	Sphcri	Subsecunda	291 nts	ERX443237	
S. lescurii	Sphle	Subsecunda	partial	ERX337183	
Takakia lepidozioides	Takle-207	Not applicable	207 nts	ERX2100030
SKQD-2076588b	
Notes.

a PHYTOZOME accession.

b 1 KP accession (Xia, Xu & Meyers, 2017).

We also analyzed the SRA database of subgenus Sphagnum (Shaw et al., 2010; Shaw et al., 2016). It was found that S. magellanicum belonging to this subgenus also encode two TAS3 loci called Sphma-285 (285 nt size) and Sphma-286 (286 nt size) (Fig. S1 and Table 1). Unlike S. fallax and S. recurvum, in S. magellanicum TAS3 loci are more similar, showing 86% identity (Fig. 2). Both Sphma-285 and Sphma-286 had 85% identity to Sphan-285 (Fig. 2). It was found that TAS3-like locus (Sphpa) from one more representative of subgenus Sphagnum (S. palustre) exhibited 98% identity to Sphma-285 (Fig. S1 and Table 1). The SRA database also contained sequence reads of two representatives from subgenus Subsecunda (Shaw et al., 2010; Shaw et al., 2016). Our BLAST analysis and subsequent assembly of retrieved reads revealed a single TAS3 locus in S. cribrosum (Sphcri, 291 nt size) showing 95% identity to Sphan-285 and 81% identity to Sphma-286 (Fig. 2, Fig. S1 and Table 1) and a partial TAS3-like sequence in S. lescurii (Fig. S1 and Table 1).

Analysis of the SRA database of Takakia lepidozioides (class Takakiopsida) allowed us to reveal only one TAS3-like sequence (Takle-207) (Fig. S1 and Table 1). The same sequence was revealed in a longer assembly which was found recently upon search of 1KP database (Xia, Xu & Meyers, 2017).

Since Takakiopsida and Sphagnopsida are most basal sister lines to all other Bryophyta (Shaw et al., 2010; Shaw, Szövényi & Shaw, 2011; Rosato et al., 2016; Puttick et al., 2018), it was very interesting to compare the structural organization of Takakiopsida and Sphagnopsida TAS3 loci with other classes of Bryophyta. Our previous detailed analysis of approximately 40 TAS3 loci in Bryophyta (Krasnikova et al., 2011; Krasnikova et al., 2013) showed that the general structure of moss TAS3 is similar in all taxa and fits the structural organization of Physcomitrella patens genes, comprising dual miR390 target sites on the 5′ and 3′ borders and internal monomeric tasiAP2 sequence followed by tasiARF sequence positioned in 20–30 bases. We revealed that phylogenetic tree of TAS3-like loci in Bryophyta showed clear subdivision of their sequences into two main clades (see Fig 5 in Krasnikova et al., 2013). The first group was formed by a cluster of sequences close to P. patens TAS3 species PpTAS3a, PpTAS3d, and PpTAS3f, and the second one—by those close to PpTAS3b, PpTAS3c, and PpTAS3e. The recent paper on the structure of TAS3 loci in lower land plants (Xia, Xu & Meyers, 2017) has shown the structure-functional basis for this phylogenetic subdivision. TAS3 species of the first group (PpTAS3a/PpTAS3d/PpTAS3f cluster) were shown to form class III of TAS3-like loci and contain, in addition to the previously reported tasiAP2 and tasiARF-a2 sequences, newly discovered tasiARF-a3 sequence positioned 5′ according to tasiAP2 (Fig. 3). Among TAS3 species of basal Bryophyta, Andreaea rupestris locus 13-Aru (Krasnikova et al., 2013) belongs to class III (Fig. 3).

Figure 3 Multiple sequence alignment of selected available nucleotide sequences of class III TAS3-like loci from Bryopsida mosses along with TAS3 locus of Folioceros fuciformis.

Alignment was generated at MAFFT6 program. The miR390 target sites are in yellow; putative tasiARF-a2 site is in green; tasiAP2 is in blue, and tasiARF-a3 is shaded. Note that all sequences are cut in the 5′-terminal TAS3 area to exclude non-aligned regions.

BLAST comparison of T. lepidozioides TAS3 with known Bryopsida loci showed that Takle-207 belongs to class II of TAS3 with typical positioning of tasiAP2 and tasiARF-a2 sequences (Fig. 2 and Fig. S1). On the other hand, none of Sphagnopsida TAS3-like sequences (Table 1) showed conventional internal structural organization of the most moss TAS3 species. The only recognizable conserved site, except miR390-targeting regions, was identified as tasiARF-a2 sequence, which was found to be conserved between two very distant TAS3 loci in S. fallax and S. recurvum (Fig. 2). The mentioned above tasiARF sequences, tasiARF-a2 and tasiARF-a3, showed no sequence similarity suggesting their independent origins. These tasiRNAs were found to be formed from different strands of the TAS3 dsRNA intermediate and target different regions of ARF genes (Xia, Xu & Meyers, 2017). Inhibition of production of both tasiARF RNAs in P. patens resulted in obvious developmental defects exhibited, in particular, as alterations in gametophore initiation, protonemal branch determinacy and caulonemal differentiation (Plavskin et al., 2016).

Figure 4 Novel potential ta-siRNA in Bryopsida plants.

(A) Multiple sequence alignments of nucleotide sequence blocks including tasiAP2 site and preceding 21 bp site of putative ta-siRNA of Andreaea rupestris TAS3-like locus WOGB_ 2010369 with the corresponding transcript sequences of moss TAS3 loci. BLASTN was used at 1 KP blast site. For the complete TAS3 transcript sequences see Xia, Xu & Meyers, 2017. The putative tasiAP2 site is in blue, and preceding putative ta-siRNA site is in violet. Andreaea1 - Andreaea rupestris WOGB_ 2010369; Andreaea2 - Andreaea_rupestris WOGB_2002765; Tetraphis1 - Tetraphis_pellucida HVBQ_2019753; Tetraphis2 - Tetraphis_pellucida HVBQ_2011866; Tetraphis3 - Tetraphis_pellucida HVBQ_2005644; Plagiomnium - Plagiomnium_insigne BGXB_2010105; Leucobryum - Leucobryum_glaucum RGKI_2062694; Racomitrium - Racomitrium_varium RDOO_2117129; Philonotis - Philonotis fontana ORKS_2058791; Dicranum - Dicranum_scoparium NGTD_2078536; Encalypta –Encalypta streptocarpa KEFD_2058811;Ceratodon - Ceratodon_purpureus FFPD_2044193; Niphotrichum - Niphotrichum_elongatum ABCD_2000143; Funaria - Funaria sp. XWHK_2042016; Schwetschkeop –Schwetschkeopsis fabronia IGUH_2166854; Aulacomnium - Aulacomnium heterostichum WNGH_2088134; Syntrichia - Syntrichia_princeps GRKU_2074985; Diphyscium1 - Diphyscium foliosum AWOI_2069791; Diphyscium2 - Diphyscium foliosum AWOI_2006305; Pohlia - Pohlia nutans GACA01023180; Bryum argenteum - Unigene33538 GCZP01053768; PHYSCO TAS3A –Physcomitrella patens TAS3a BK005825. (B) The example target transcript sequence (lower case letters) from Leucobryum albidum (VMXJ_2127900) is presented alongside with the predicted novel ta-siRNA shown in blue and above the transcript. Lower case letters in ta-siRNA indicate non-Watson-Crick pairing positions. Complementary mRNA sequences are in violet; conserved amino acid sequence signatures are in yellow. Numbers indicate codon positions of the target gene.

Comparison of nucleotide sequences between TAS3 species of several moss classes revealed in many plants obvious similarity of nucleotide sequence blocks including tasiAP2 site and immediate upstream 21 bp block occurring in the same 21-bp-phase (Fig. 4A). We hypothesized that this sequence block may correspond to novel previously unrecognized ta-siRNA in many moss species. This putative siRNA in its single-stranded form really presents in P. patens transcriptome as minus-sense 21-nucleotide ta-siRNA (see NCBI SRA accessions SRX903096–SRX903105) like tasiARF RNA (Arif et al., 2012). Thus, we speculated that novel hypothetical ta-siRNA might be produced from TAS3, and its minus-strand is complementary to uncharacterized well-conserved, protein-coding moss mRNAs which have homologs also in conifers and angiosperms (Fig. 4B; Fig. S2).

TAS3 loci in Anthocerotophyta

Taking into account the finding of TAS3-like loci in classes Sphagnopsida and Takakiopsida and previously published data (Krasnikova et al., 2013; Xia, Xu & Meyers, 2017), one can conclude that the only remaining blind-spot in land plants with respect to TAS3 is represented by phylum Anthocerotophyta. Relationships between liverworts, mosses and hornworts are still obscure. Moreover, the question remains which bryophyte phylum is a sister line to all other land plants (Qiu, 2008; Shaw, Szövényi & Shaw, 2011; Harrison, 2017; Puttick et al., 2018). Recent molecular phylogenetic analysis, in which three bryophyte lineages were resolved, revealed that a clade with mosses and liverworts could form a sister group to the tracheophytes, whereas the hornworts is sister line to all other land plants (Wickett et al., 2014). However, analyses of the plastid genome sequences suggested another branching order of the phylogenetic tree, with hornworts rather than moss/liverwort clade being a sister group to tracheophytes (Lewis, Mishler & Vilgalys, 1997; Samigullin et al., 2002; Ruhfel et al., 2014; Lemieux, Otis & Turmel, 2016). Moreover, some very recent nuclear gene comparisons also suggested that hornworts could be a sister clade to tracheophytes, and liverworts plus mosses might be closer to a common ancestor of land plants (Rosato et al., 2016; Bowman et al., 2017). However, this ancestor could have more tracheophyte-like characteristics than Setaphyta (liverworts/mosses) due to secondary simplification in liverworts (Puttick et al., 2018).

Analysis of the SRA database of Anthocerotophyta revealed a putative TAS3-like sequence in Folioceros fuciformis (family Anthocerotaceae). Unexpectedly, the discovered TAS3-like sequence (Folfu) was found to be 244 nucleotides in length and obviously similar to Bryophyta class III TAS3 species (Fig. 3, Fig. S3 and Table 2). The identity of Folfu to some moss TAS3 sequences exceeds 80% being therefore even higher than between some related Bryopsida species (Fig. 3). Thus these data clearly indicate a close relation of TAS3 in Anthocerotophyta to Bryophyta TAS3 (excepting Sphagnopsida).

TAS3 loci in Marchantiophyta

Some of the recent molecular phylogenetic reconstructions suggested that Marchantiophyta species could represent a sister clade to all other land plants (see above). Therefore, finding and comparative analyses of TAS3 loci in this taxon represented a significant interest for understanding early events in TAS3 evolution. In contrast to class Marchantiopsida, where putative TAS3 and pre-miR390 loci were previously identified (Krasnikova et al., 2013; Lin et al., 2016; Tsuzuki et al., 2016), for class Jungermanniopsida only potential pre-miR390 loci were found in Pellia endiviifolia and Harpanthus flotovianus (Krasnikova et al., 2013; Alaba et al., 2015). Assuming that miR390 was found to be among eight most conserved miRNA species in land plants (Xia et al., 2013; You et al., 2017; Liu et al., 2018), Jungermanniopsida could be expected to encode TAS3 loci.

Table 2 List of the putative TAS3 loci in Anthocerotophyta and Marchantiophyta.

Plant species	Class/subclass	Order	Length	Sequence source	
Folioceros fuciformis	Anthocerotopsida/Anthocerotidae	Anthocerotales	244 nts	SRS2162762	
Marchantia polymorpha
1-Mpo	Marchantiopsida/Marchantiidae	Marchantiales	256 nts	KC812742	
Marchantia emarginata	Marchantiopsida/Marchantiidae	Marchantiales	262 nts	SRX1952816	
Conocephalum japonicum	Marchantiopsida/Marchantiidae	Marchantiales	252 nts	SRX1952810	
Ricciocarpos natans	Marchantiopsida/Marchantiidae	Marchantiales	235 nts	ERX337127	
Dumortiera hirsuta	Marchantiopsida/Marchantiidae	Marchantiales	243 nts	SRX1126014	
Plagiochasma appendiculatum	Marchantiopsida/Marchantiidae	Marchantiales	247 nts	SRX1741567	
Conocephalum conicum	Marchantiopsida/Marchantiidae	Marchantiales	248 nts	ILBQ_2006554a	
Lunularia cruciata	Marchantiopsida/Marchantiidae	Lunulariales	220 nts	TXVB_2071521a	
Marchantia paleaceae	Marchantiopsida/Marchantiidae	Marchantiales	257 nts	HMHL_2051051a	
Metzgeria crassipilis	Jungermanniopsida/Metzgeriidae	Metzgeriales	226 nts	ERX337128	
Pellia endiviifolia	Jungermanniopsida/Pelliidae	Pelliales	192 nts	SRX726500	
Notes.

a 1 KP accession (Xia, Xu & Meyers, 2017).

To detect new potential TAS3 loci, we performed BLAST analysis of the SRA database for species of class Jungermanniopsida using Marchantia polymorpha TAS3 sequence (1-Mpo) as a query. Using this approach we revealed a set of reads and assembled a single TAS3-like locus (Pelen-192) for Pellia endiviifolia (192 nt size). In addition, TAS3 locus of 226 nucleotides in length was found in Metzgeria crassipilis (Metcr-226) (Fig. 5, Table 2, Fig. S3). The latter locus was also recently revealed in a search of 1KP database (Xia, Xu & Meyers, 2017).

Figure 5 Analysis of TAS3 loci in Marchantiophyta plants.

(A) Multiple sequence alignment of available nucleotide sequences of TAS3-like loci from Marchantiophyta plants along with TAS3 loci of Takakia lepidozioides and Folioceros fuciformis. Alignment was generated at MAFFT6 program. The miR390 target sites are in yellow, and putative tasiARF-a2 site is in green; tasiAP2 site is in blue. (B) The minimal evolution phylogenetic tree based on analysis of the aligned TAS3 genes from Marchantiophyta plants. This tree was generated according MAFFT6 program. For full plant names and accession numbers see Table 2.

TAS3 1-Mpo sequence was further used for BLAST analysis of other Marchantiopsida sequences available at the NCBI SRA database. As a result, we retrieved sequence reads and assembled five full-length TAS3-like sequences in Plagiochasma appendiculatum (Plaap-247), Dumortiera hirsuta (Dumhi-243), Marchantia emarginata (Marem-262), Ricciocarpos natans (Ricna-235) and Conocephalum japonicum (Conja-252) (Fig. 5, Table 2, Fig. S3). Recent bioinformatics analysis of 1KP database revealed three additional full-length TAS3-like sequences in Conocephalum conicum, Lunularia cruciata and Marchantia paleaceae (Xia, Xu & Meyers, 2017) (Table 2). Thus, totally 11 TAS3-like loci have been found in Marchantiophyta.

Figure 6 Pairwise sequence comparisons of selected nucleotide sequences of TAS6/TAS3-like loci from mosses with TAS6/TAS3 of Physcomitrella patens precursor RNA (accession JN674513).

BLASTN was used at 1 KP blast site. The miR390 target sites are in yellow; putative miR156/miR529 sites are underlined; tasiAP2 is in blue; putative tasiARF-a2 site is in green; tasiARF-a3 is shaded.

TAS6 loci in Bryophyta

Previous studies of P. patens revealed three novel non-coding PHAS loci (TAS6) which were located in rather close genomic proximity to PpTAS3 loci (PpTAS3a, PpTAS3d, and PpTAS3f) and expressed as common RNA precursors with these TAS3 species (Cho, Coruh & Axtell, 2012; Arif et al., 2012; Arif, Frank & Khraiwesh, 2013). Moreover, miR529 and miR156 were suggested to influence accumulation of ta-siRNAs specific not only for TAS6, but also for PpTAS3a (Cho, Coruh & Axtell, 2012). We have found that localization of TAS6 loci close to TAS3 genes in common transcripts was not unique for P. patens (subclass Funariidae), since these loci were also found to be encoded by three other mosses of subclasses Bryidae and Dicranidae (Krasnikova et al., 2013).

Table 3 List of the putative TAS6/TAS3 loci of Bryophyta in transcribed sequences found in 1 KP database.

Plant species	Class/subclass	Order	Lengtha and type	Sequence source	
Timmia austriaca	Bryopsida/Timmiidae	Timmiales	TAS6/TAS3 (874 nts)	ZQRI-2061439
ZQRI-2063082	
Thuidium delicatulum	Bryopsida/Bryidae	Hypnales	TAS6/TAS3 (837 nts)	EEMJ-2003175	
Hypnum subimponens	Bryopsida/Bryidae	Hypnales	TAS6/TAS3 (823 nts)	LNSF-2068452	
Pseudotaxiphyllum elegans	Bryopsida/Bryidae	Hypnales	TAS6/TAS3 (1,590 nts)	QKQO-2009669	
Anomodon attenuatus	Bryopsida/Bryidae	Hypnales	TAS6/TAS3 (843 nts)	QMWB-2059873	
Anomodon rostratus	Bryopsida/Bryidae	Hypnales	TAS6/TAS3 (829 nts)	VBMM-2003482	
Schwetschkeopsis fabronia	Bryopsida/Bryidae	Hypnales	TAS6/TAS3 (854 nts)	IGUH-2166854	
Leucodon sciuroides	Bryopsida/Bryidae	Hypnales	TAS6/TAS3 (852 nts)	ZACW-2016434	
Fontinalis antipyretica	Bryopsida/Bryidae	Hypnales	TAS6/TAS3 (1,410 nts)	DHWX-2007057	
Rhytidiadelphus loreus	Bryopsida/Bryidae	Hypnales	TAS6/TAS3 (830 nts)	WSPM-2009782	
Rhynchostegium serrulatum	Bryopsida/Bryidae	Hypnales	TAS6/TAS3 (853 nts)	JADL-2047695	
Climacium dendroides	Bryopsida/Bryidae	Hypnales	TAS6/TAS3 (809 nts)	MIRS-2012325	
Calliergon cordifolium	Bryopsida/Bryidae	Hypnales	TAS6 (95 nts)	TAVP-2006322	
Neckera douglasii	Bryopsida/Bryidae	Hypnales	TAS6/TAS3 (839 nts)	TMAJ-2023603	
Plagiomnium insigne	Bryopsida/Bryidae	Bryales	TAS6/TAS3 (914 nts)	BGXB-2010105	
Orthotrichum lyellii	Bryopsida/Bryidae	Orthotrichales	TAS6 (192 nts)	CMEQ-2080784	
Hedwigia ciliata	Bryopsida/Bryidae	Hedwigiales	TAS6/TAS3 (877 nts)	YWNF-2050742	
Philonotis fontana	Bryopsida/Bryidae	Bartramiales	TAS6/TAS3 (893 nts)	ORKS-2058791	
Aulacomnium heterostichum	Bryopsida/Bryidae	Rhizogoniales	TAS6/TAS3 (863 nts)	WNGH-2088134	
Scouleria aquatic	Bryopsida/Dicranidae	Scouleriales	TAS6/TAS3 (partial)	BPSG-2088977	
Syntrichia princeps	Bryopsida/Dicranidae	Pottiales	TAS6/TAS3 (partial)	GRKU-2074985	
Leucobryum glaucum	Bryopsida/Dicranidae	Dicranales	TAS6/TAS3 (763 nts)	RGKI-2062694	
Leucobryum albidum	Bryopsida/Dicranidae	Dicranales	TAS6/TAS3 (763 nts)	VMXJ-2128109	
Dicranum scoparium	Bryopsida/Dicranidae	Dicranales	TAS6 (105 nts)	NGTD-2092412	
Ceratodon purpureus	Bryopsida/Dicranidae	Pseudoditrichales	TAS6/TAS3 (1,121 nts)	FFPD-2005850
SRX2065999	
Racomitrium varium	Bryopsida/Dicranidae	Grimmiales	TAS6/TAS3 (724 nts)	RDOO-2117129	
Physcomitrium_sp.	Bryopsida/Funariidae	Funariales	TAS6 (partial)	YEPO-2071108	
Physcomitrium_sp.	Bryopsida/Funariidae	Funariales	TAS6 (178 nts)	YEPO-2000016	
Physcomitrium_sp.	Bryopsida/Funariidae	Funariales	TAS6/TAS3 (821 nts)	YEPO-2016361	
Encalypta streptocarpa	Bryopsida/Funariidae	Encalyptales	TAS6/TAS3 (883 nts)	KEFD-2058811	
Diphyscium foliosum	Bryopsida/Diphysciidae	Diphyscales	TAS6/TAS3 (832 nts)	AWOI-2069791	
Tetraphis pellucida	Tetraphidopsida	Tetraphidales	TAS6 (partial)	HVBQ-2112923	
Atrichum angustatum	Polytrichopsida	Polytrichales	TAS6/TAS3 (810 nts)	ZTHV-2082998	
Andreaea rupestris	Andreaeopsida	Andreaeales	TAS6/TAS3 (869 nts)	WOGB-2010369	
Takakia lepidozioides	Takakiopsida	Takakiales	TAS6/TAS3 (1,040 nts)	SKQD-2076588	
Notes.

a The length indicates total size of TAS6-TAS3 complex element (from the 5′ miR529 target site in TAS6 to 3′ miR390 target site in TAS3) or isolated TAS6 (between miR529 and miR156 target sites).

For further search of the combined TAS6/TAS3 loci, we performed bioinformatics analysis of 1KP database. Although nucleotide sequences of miR156 and related miR529, as well as their recognition sites in RNA transcripts, are highly conserved among land plants (Morea et al., 2016; Axtell & Meyers, 2018), the internal sequences between dual miR156/miR529 recognition sites show little or no similarity even between different TAS6 loci of P. patens (Arif et al., 2012). So we used, as queries for BLAST search, the individual full-length TAS6/TAS3 loci including most characterized locus encoding PpTAS3a (Fig. 6), as well as those for PpTAS3d and PpTAS3f. First, it was found that in addition to four previously found Bryopsida species, encoding TAS6/TAS3 loci, these loci could be revealed in basal subclasses Timmiidae (Timmia austriaca) and Diphysciidae (Diphyscium foliosum) (Shaw, Szövényi & Shaw, 2011; Table 3, Fig. S4). List of TAS6/TAS3 loci in other moss subclasses was also significantly extended: we found 18 new loci in Bryidae, seven loci in Dicranidae and four loci in Funariidae (Table 3, Fig. S4). These novel loci showed recognizable but varying sequence similarities to the PpTAS3a-containing locus (Fig. 6). Second, most importantly, putative TAS6/TAS3 loci were revealed in four basal classes of Bryophyta, namely, Tetraphidiopsida, Polytrichopsida, Andreaeopsida and Takakiopsida (Table 3, Fig. S4). These novel loci had a similar organization to Bryopsida TAS6/TAS3 species (Fig. 6). However, no TAS6-specific sequence signatures were found in the vicinity of genomic S. fallax and M. polymorpha TAS3 loci upon analysis of the corresponding Phytozome genome contigs.

Phylogeny of SGS3 as a characteristic molecular component of TAS3 pathway

It was shown that some species of green algae could encode ancient types of dicer-like proteins, RDRs, and AGOs. On the other hand, no encoded SGS3 proteins were revealed for these algae (Zheng et al., 2015). Since SGS3 was found to be essential for production of tasiARF RNAs in moss P. patens (Plavskin et al., 2016), we performed sequence analyses to identify possible SGS3 genes in charophytes. For identification of SGS3 protein orthologs among land nonvascular plants and charophytes, we used as a query the most conserved region of P. patens SGS3 including short zinc binding zf-XS domain and RNA recognition XS domain (Bateman, 2002; Zhang & Trudeau, 2008). Importantly, the short N-terminal zf-XS domain is characteristic for functional SGS3 proteins, since the XS domain-containing protein of Selaginella moellendofii lacking TAS-generating machinery (Banks et al., 2011) possesses no zf-XS domain upstream of XS domain and instead contains the C-terminal RING zf region (see NCBI accession XP_002979112). However, it should be noted that the lack of TAS3 pathway and SGS3 is not universal for lycophytes (Xia, Xu & Meyers, 2017).

Figure 7 The phylogenetic tree constructed from conserved regions of SGS3 protein sequences from 27 selected streptophytes by the Neighbor-Joining method with 1000 bootstrap replications.

There were a total of 419 positions in the final dataset. Evolutionary analyses were conducted in MEGA7 (Kumar, Stecher & Tamura, 2016). Bootstrap support values ≥50% are shown. The evolutionary distances were computed using the JTT matrix-based method and are in the units of the number of amino acid substitutions per site. The rate variation among sites was modeled with a gamma distribution (shape parameter = 1). The tree was rooted at two Klebsormidiaceae charopytes. Accession numbers from NCBI ot PHYTOZOME data banks see in Fig. S5.

In addition to class Bryopsida, SGS3 protein sequences were revealed for members of classes Marchantiopsida, Jungermanniopsida, Anthocerotopsida, Takakiopsida and Sphagnopsida (Fig. 7 and Fig. S5). Most importantly, search for the SGS3 coding sequences in transcriptomes of four charophyte classes (Klebsormidiophyceae) also revealed the SGS3-like proteins in representatives of all these taxa (Fig. 7, Figs. S5, S6). This observation was in agreement with the fact that SGS3-like coding sequence was found in the fully sequenced and annotated genome of Klebsormidium nitens (NCBI accession GAQ92898) (Hori et al., 2014). Moreover, the characteristic motifs of land plant SGS3 proteins (Bateman, 2002) were revealed in the protein sequences from charophyte algae (Figs. S5, S6).

Importantly, in the dendrogram based on comparisons of 27 aligned SGS3 protein sequences, the position of charophytes (Fig. 7) nearly corresponded to the commonly accepted Viridiplantae phylogenetic tree (Shaw, Szövényi & Shaw, 2011; Delwiche & Cooper, 2015; Harrison, 2017), where class Zygnemophyceae (Spirogyra pratensis) was a sister group for all land plants. It has become clear that evolving the SGS3-like genes was not directly connected to the appearance of TAS loci in Viridiplantae, since Chlorophyta species, lacking SGS3, encode not only critical enzyme machinery including DCLs, RDRs, and AGOs (You et al., 2017), but also PHAS loci (Zheng et al., 2015). Despite our extensive searches, no SGS3 genes could be identified also in brown and red algae, and this is in agreement with previously published data on green algae (Zheng et al., 2015).

Discussion

Our current analyses revealed previously undiscovered TAS3 loci in bryophytes from classes Takakiopsida, Sphagnopsida and Anthocerotopsida. In Folioceros fuciformis (family Anthocerotaceae) we found a TAS3-like sequence which is obviously similar to Bryopsida class III TAS3 species (Fig. 3), whereas Takakiopsida TAS3 locus showed relatedness to class II of TAS3 with typical positioning of tasiAP2 and tasiARF-a2 sequences and no tasiARF-a3 sites (Xia, Xu & Meyers, 2017). Unexpectedly, all predicted Sphagnopsida TAS3 loci showed no conventional internal structural organization of the moss class II and class III TAS3 species. Excepting miR390-targeting regions, the only recognizable conserved site was tasiARF-a2 sequence (Fig. 2).

It was shown that structural organization of Marchantiopsida and Jungermanniopsida TAS3 loci were quite similar, whereas Marchantiophyta TAS3 species were obviously different from those of Bryophyta. These loci were proposed to belong to TAS3 class I with species containing two conserved sequence blocks which presumably represent functional ta-siRNAs (Xia, Xu & Meyers, 2017). One of these blocks was found in the vicinity of the 3′-terminal miR390 binding site and corresponded to Bryopsida tasi-AP2 sequence (Krasnikova et al., 2013), whereas another one (tasiARF-a1), unique among lower land plants, was located closer to the 5′-terminal miR390 binding site in Marchantiopsida and Jungermanniopsida TAS3 (Tsuzuki et al., 2016; Xia, Xu & Meyers, 2017) (Fig. 5, Fig. S3).

Assuming occurrence of tasiARFs as potential products of TAS3 in all main lineages of land plants (Marchantiophyta, Anthocerotophyta and Bryophyta), recent paper proposed that the earliest function of TAS3 could contribute to the production of ta-siRNAs targeting ARF genes, and, since green algae encode no ARF genes, TAS3 likely appeared first in land plants (Xia, Xu & Meyers, 2017). However, extensive comparative sequence analysis showed that charophyte algae representing the sister group to all land plants (colonized terrestrial environments approximately 480 million years ago, see for references Fischer (2018) and Puttick et al. (2018)) could also encode ARF-like proteins including all sequence domains typical for bryophyte and angiosperm ARFs (Mutte et al., 2017). Moreover, our current data showed that TAS3-like loci are encoded by the representatives of all main taxa among non-vascular plants. These observations suggest that the TAS3 evolution started in a common ancestor of land plants, likely belonging to a still unknown lineage of charophytes. Identification of the canonical motifs of land plant SGS3 in charophyte proteins (see above) indirectly supports this speculation. However, it should be kept in mind that evolving the SGS3-like genes could not be connected solely to the appearance of PHAS loci in Viridiplantae, since green algae and brown algae species were found to encode not only essential silencing machinery enzymes including DCLs, RDRs and AGOs, but also PHAS loci (Billoud et al., 2014; Zheng et al., 2015; Singh et al., 2015; Dueck et al., 2016; You et al., 2017; Cock et al., 2017). Finally, it can be proposed that the failure to identify charophyte TAS3 loci may be related to (i) the incompleteness of the available sequence data; (ii) evolving by charophytes the one-hit TAS3 genes (De Felippes et al., 2017); or (iii) the use of miRNA species with sequences other than land plant miR390 for TAS precursor processing.

Conclusions

The current data on the structural organization of TAS3-like loci in all main classes of land non-vascular plants reveal three types of TAS3 loci, namely, (i) Bryopsida-like TAS3 (classes II and III, Xia, Xu & Meyers, 2017) found in Bryophyta plants (excepting Sphagnopsida) and Anthocerotopsida, (ii) Marchantiophyta-like TAS3 (class I, Xia, Xu & Meyers, 2017) and (iii) Sphagnopsida-like TAS3 (this paper). Clearly recognizable common ta-siRNAs is represented in these TAS3 types by tasiARF sequences. Occurrence of primitive SGS3 and ARF genes in charophytes (Mutte et al., 2017 and this paper) supports the idea that TAS3-like genes first appeared in the hypothetical common precursor of land plants (Xia, Xu & Meyers, 2017) to regulate auxin signaling (Leyser, 2018).

Supplemental Information

Figure S1 Nucleotide sequences of TAS3 loci in mosses of Sphagnum and Takakia genera

Click here for additional data file.

Figure S2 Putative ta-siRNA preceding tasiAP2 site in the same phase

Click here for additional data file.

Figure S3 Nucleotide sequences of TAS3 loci in plants from Anthocerotophyta and Marchantiophyta

Click here for additional data file.

Figure S4 RNA transcripts of the selected Bryophyta TAS6/TAS3 loci found in 1 KP database

Click here for additional data file.

Figure S5 Conserved portions of SGS3 amino acid sequences used for constructing of phylogenetic tree

Click here for additional data file.

Figure S6 SGS3 amino acid and nucleotide sequences in charophytes

Click here for additional data file.

Supplemental Information 1 Sequences reported in this paper

Click here for additional data file.

We thank researchers who contributed samples used in this study to the 1KP initiative.

Additional Information and Declarations

Competing Interests

Author Contributions

DNA Deposition

The authors declare there are no competing interests.

Sergey Y. Morozov conceived and designed the experiments, analyzed the data, prepared figures and/or tables, authored or reviewed drafts of the paper, approved the final draft.

Irina A. Milyutina conceived and designed the experiments, performed the experiments, contributed reagents/materials/analysis tools, approved the final draft.

Tatiana N. Erokhina and Liudmila V. Ozerova performed the experiments, contributed reagents/materials/analysis tools, prepared figures and/or tables, approved the final draft.

Alexey V. Troitsky and Andrey G. Solovyev conceived and designed the experiments, analyzed the data, authored or reviewed drafts of the paper, approved the final draft.

The following information was supplied regarding the deposition of DNA sequences:

New sequences described here are accessible via GenBank accession numbers MF682529 and MF682530.

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
