# Peer review of "TAS3 miR390-dependent loci in non-vascular land plants: towards a comprehensive reconstruction of the gene evolutionary history"

_PeerJ, doi:10.7717/peerj.4636_

## Round 0.1 · original submission · Minor Revisions

Please follow the suggestions of the reviewers and also see the annatayed manuscript

·

Basic reporting

1. While main figures with numerous BLAST-generated sequence comparisons of TAS3-like loci correspond to standards in description ta-si RNA motives, this monotonous illustrative material insufficiently emphasizes author’s main findings. I advise to summarize sequence comparison data in a more general scheme, illustrating common TAS3-motives and placing the information on a general phylogenetic context with the help of available tools like MAFFT https://mafft.cbrc.jp/alignment/server/.
2. Although English is not my native language, I feel it is fine. Typos are absent. The paragraph text between lines 220-226 requires more clear narration, I advise to rewrite it.

Experimental design

3. Fig. 10 shows phylogenetic tree of SGS3 proteins. I suggest to use more sequences for four basal members (Charophyte algae) to create more informative dendrogram.

Validity of the findings

4. Fig. S2 shows nucleotide sequences and encoded amino acid sequences of moss mRNAs potentially recognized by the putative ta-siRNA preceding in the same phase tasiAP2 site in some moss TAS3 loci. However, the authors did not discuss possible functions of these moss proteins and their similarities with other eukaryotic polypeptides.

Additional comments

The article by Morozov et al traces the evolutionary history of two loci encoding small interfering RNAs (TAS3 and TAS6) in mosses. The TAS3 and TAS6 target mRNAs for ARF transcription factor genes and are deemed to be important for the growth and development of land plants. Processing of TAS3/6 precursor is mediated by evolutionary conserved mir390 microRNA. So, identification of TAS3/mir390 ancestors in deep evolutionary lineages provides useful information to clarify ambiguities in modern concepts of evolution of land plants. Author took advantage of the vast amount of available NGS genomic and transcriptomic data to search for putative TAS3-like loci in basal lineages in Bryophyta sensu lato. Using both bioinformatics and experimental approaches they identified novel putative TAS3-like loci in Sphagnopsida and Takakiopsida. By comparing the structural organization of novel loci with those previously identified in other moss lineages authors subdivided them into two major classes and found a new additional sequence element, potentially corresponding to novel moss tasiRNA. Inter-division comparison revealed striking similarity of sequences from Folioceros fuciformis (a hornwort) and Bryopsida, and, in contrast, clear differences between TAS3-like loci in Bryophyta and liverworts. Another interesting finding reported by the author is the discovery of additional “combined” TAS6/TAS3 loci, specifying long RNA precursors for these tasiRNAs in moss transcriptome databases. Finally, in search of the ancestors of the components of tasiRNA generating machinery, authors found genes encoding putative orthologs of the so-called SGS3-proteins in non-vascular terrestrial plant and charophytes, but not in brown and red algae, confirming current view on the evolution of RNA silencing machinery in plants.
Altogether, I consider the data presented in the submitted manuscript as an important step forward in our understanding of the evolutionary pathway of miR390-dependent TAS3 loci in land plants. The article will be useful to readers of “PeerJ ” interested in phylogeny of mosses, tasiRNA biology, plant evolutionary genomics.
The article may be published in the form close to original one after several minor revisions.

Reviewer 2 ·

Basic reporting

In this study, Morozov and co-workers used a PCR-based approach combined bioinformatic analysis of publicly available genome and transcriptome resources to identify (i) new TAS3- and TAS6-like loci encoding trans-acting siRNAs that target ARF transcription factor genes (tasiARFs) and other target genes in basal groups of non-vascular land plants (mosses, liverworts and hornworts) as well as SGS3-like genes in charophyte algae, the sister group to all land plants. These findings extend previous findings of Morozov's and Meyers' labs and imply that the tasiARF-generating machinery may have evolved in a common ancestor of land plants. The manuscript is written in clear and professional English (minor suggestions to improve the text are provided in the attached PDF file). All the relevant background and context literature is cited and discussed. The main conclusions are well justified. The only major revision that could be recommended concerns the Figures 1-4 and 6-9 that show pairwise alignments of TAS loci. I would encourage the authors to make those Figures more compact (some of them span more than one or even two pages) and, instead of several pairwise alignments, show one multiple alignment in each of the Figures and, in addition to color-coding, indicate/name the two mR390 binding sites, and each of the two or three distinct conserved tasiRNA species. The multiple alignment should help to easier evaluate the conservation of tasiRNA species in different plants, especially in their 5'-seed region and positions 10 and 11, which are important for tasiRNA-directed cleavage of target gene transcripts. Schematic representation of a TAS gene locus above the alignments in each of those Figures would also help to grasp the differences in structural organizations.

Experimental design

The experimental PCR approach and bioinformatic analyses are described in sufficient detail. The study meets high technical and ethical standards.

Validity of the findings

The findings are valid and important for the fields of RNA silencing and gene regulation in plants and of plant evolution.

Additional comments

In this study, Morozov and co-workers used a PCR-based approach combined bioinformatic analysis of publicly available genome and transcriptome resources to identify (i) new TAS3- and TAS6-like loci encoding trans-acting siRNAs that target ARF transcription factor genes (tasiARFs) and other target genes in basal groups of non-vascular land plants (mosses, liverworts and hornworts) as well as SGS3-like genes in charophyte algae, the sister group to all land plants. These findings extend previous findings of Morozov's and Meyers' labs and imply that the tasiARF-generating machinery may have evolved in a common ancestor of land plants. The manuscript is written in clear and professional English (minor suggestions to improve the text are provided in the attached PDF file). All the relevant background and context literature is cited and discussed. The main conclusions are well justified. The only major revision that could be recommended concerns the Figures 1-4 and 6-9 that show pairwise alignments of TAS loci. I would encourage the authors to make those Figures more compact (some of them span more than one or even two pages) and, instead of several pairwise alignments, show one multiple alignment in each of the Figures and, in addition to color-coding, indicate/name the two mR390 binding sites, and each of the two or three distinct conserved tasiRNA species. The multiple alignment should help to easier evaluate the conservation of tasiRNA species in different plants, especially in their 5'-seed region and positions 10 and 11, which are important for tasiRNA-directed cleavage of target gene transcripts. Schematic representation of a TAS gene locus above the alignments in each of those Figures would also help to grasp the differences in structural organizations.

Annotated reviews are not available for download in order to protect the identity of reviewers who chose to remain anonymous.

---

## Round 0.2 · Minor Revisions

Dear Sergey,

Both reviewer suggest to accept your paper, but I have noticed that your Figure 6 does not correspond to the legend. Can you please either replace Figure 6 or its legend before the paper can be accepted?

·

Basic reporting

Suggested revisions were made, figures were improved, addressed issues clarified.

Experimental design

New multiple alignments significantly improve perception of the phylogenetic data

Validity of the findings

The author's attempt to reconstrut the evolutionary history of TAS3 miR390-dependent loci in non-vascular land plants shold be considered interesting and convincing.

Additional comments

I am fully satisfied by the amendments and revisions made and support publication of the revised form of this manuscript

Reviewer 2 ·

Basic reporting

In the revised manuscript the authors have addressed all my critical points.

Experimental design

The design meets high stndards

Validity of the findings

The findings are valid.

Additional comments

In the revised manuscript the authors have addressed all my critical points.

---

## Round 0.3 · accepted · Accept

Thank you for working with the reviewers to improve your paper.

#